# Real-World Data Confirm That the Integration of Deuterium Depletion into Conventional Cancer Therapy Multiplies the Survival Probability of Patients

**DOI:** 10.3390/biomedicines13040876

**Published:** 2025-04-04

**Authors:** Gábor Somlyai, András Papp, Ildikó Somlyai, Beáta Zs Kovács, Mária Debrődi

**Affiliations:** 1HYD LLC for Cancer Research and Drug Development, Villányi út 97, 1118 Budapest, Hungarybzskovacs@hyd.hu (B.Z.K.); drdebrodi@gmail.com (M.D.); 2Department of Public Health, Faculty of Medicine, University of Szeged, 6720 Szeged, Hungary; papp57andras@freemail.hu

**Keywords:** deuterium, deuterium-depleted water, median survival time, cancer, sub-molecular regulatory system

## Abstract

**Background**: Over thirty years of basic research has demonstrated that the deuterium-to-hydrogen ratio plays a pivotal role in regulating metabolism and cell growth via a sub-molecular regulatory system that orchestrates the intricate complexity of life in eukaryotic organisms. Deuterium depletion, achieved through deuterium-depleted water (DDW), has shown anticancer effects in vitro, in vivo, and in Phase 2 prospective and retrospective clinical studies. **Methods**: In this population-based observational study, 2649 cancer patients undergoing conventional therapy and consuming DDW were included between October 1992 and October 2024. With various cancer types and stages and conventional therapies received, they are representing a broad spectrum of the Hungarian cancer population. Survival was selected as the primary endpoint, and the median survival time (MST) of these patients and various subgroups was calculated and compared to the overall Hungarian cancer population’s MST of 2.4 years. **Results**: For the entire study population, MST from diagnosis was 12.4 years (95% CI: 9.8–14.9), and from the initiation of DDW treatment, 7.6 years (95% CI: 5.9–9.3). **Conclusions**: Utilizing DDW enables targeted intervention in the sub-molecular regulatory system, paving the way for innovative therapeutic applications and a more profound understanding of cellular processes. Integrating deuterium depletion into conventional cancer therapies has the potential to significantly enhance survival rates and reduce cancer-related mortality by 75–80%.

## 1. Introduction

Malignant neoplasms are among the leading causes of death worldwide. In individuals aged 45 to 64 years, cancers cause the majority of deaths [1]. In Europe, there were about 2.74 million new cases [2] in 2022 and 1.1 million cancer-related deaths in 2021 [3]. The United States also faced a staggering 1.9 million new cancer diagnoses and over 600,000 cancer-related deaths [4,5]. In 2021, Hungary reported approximately 70,000 new cancer cases and 33,000 deaths attributed to cancer [6]. A recent study projected a rise globally in the number of cancer cases in men from 10.3 million in 2022 to 19 million by 2050 (>84% increase). Deaths are expected to increase from 5.4 million to 10.5 million (>93% increase), with a more than twofold rise (>117%) projected among men aged 65 and older [7].

Despite the dedicated efforts of scientists and the pharmaceutical industry to develop safer and more effective cancer treatments, the global impact remains limited. Between 2000 and 2016, the FDA approved 92 novel cancer drugs for 100 indications based on 127 clinical trials. However, these drugs’ median absolute survival benefit was only 2.4 months [8]. This stark reality underscores the need for transformative advancements in cancer research and treatment. It prompts a critical reassessment of the current strategy, mainly the decades-long focus on identifying new mutations and targeting elements of signaling pathways. While this approach has dominated cancer research for over 40 years, its overall efficacy in overcoming the disease is now being questioned. The challenge of cancer remains profound, and a definitive solution is yet to be found.

Recently, two comprehensive review papers summarized the outcomes of three decades of research on deuterium (D) and deuterium-depleted water (DDW) [9,10]. These studies—initiated by pioneering research published in 1993 that first demonstrated the cell growth inhibiting effect of DDW and proved to be a significant milestone in exploring alternative cancer therapies [11]—confirmed the anticancer effects of deuterium depletion through multiple independent experiments, elucidating the underlying mechanisms and introducing a novel concept and target for cancer drug development. It has been demonstrated that replacing regular water (with 16 mmol/L D concentration, equivalent to 150 ppm, as found in natural waters) with deuterium-depleted water (3.2 mmol/L D concentration, equivalent to 30 ppm) inhibits cell growth in vitro and leads to complete tumor regression in vivo. Conversely, research has also shown that deuterium-enriched water, containing deuterium in 2–4-fold concentrations above natural levels (300–600 ppm), stimulates cell growth.

These findings, combined with earlier results [12,13,14] indicating a correlation between increased intracellular pH and cell division, suggest a critical role for the Na⁺/H⁺ exchanger in this process. It was supposed that the exchanger, sitting in the cell membrane and activated when a growth hormone binds to its receptors, preferentially transfers the lighter hydrogen isotope (protium), leading to an increased intracellular deuterium-to-hydrogen (D/H) ratio. Yeast ATPase was shown to prefer only hydrogen (not deuterium) as a substrate in proton transfer [15] and in fibroblasts expressing yeast ATPase in their membrane, tumorigenicity and increased intracellular pH was observed [14]. This D/H ratio is proposed to act as a key sub-molecular signal for initiating cell growth, which is supported by the data that deuterium-enriched media stimulated cell growth [11]. 

Since Warburg’s discovery [16], it has been well established that a hallmark of most cancer cells is their reliance on anaerobic metabolism, even in the presence of oxygen [16,17]. It has been hypothesized that properly functioning mitochondria in healthy cells produce deuterium-depleted metabolic water, which inhibits cell growth by counteracting the increase of the D/H ratio caused by activation of the Na⁺/H⁺ exchanger. When food is metabolized into water and carbon dioxide, the resulting metabolic water is deuterium-depleted, with a degree of depletion depending on the composition of the food. This hypothesis is supported by the dissimilar D concentrations in macronutrients: in proteins 135 ppm, slightly reduced vs. natural waters; significantly reduced, 109 to 118 ppm, in lipids; but about 150 ppm in carbohydrates [18]. This metabolic water thus helps maintain a lower D/H ratio, acting as a regulatory mechanism to reduce the likelihood of reaching the threshold D/H ratio required to trigger cell growth. 

Supporting this hypothesis, recent experiments demonstrated that feeding tumor-bearing mice with deuterium-depleted yolk [18] but with normal water extended their survival, underlining the critical role of mitochondrial activity in controlling the D/H ratio and inhibiting tumor progression. Consequently, in line with Warburg’s idea, cancer cells’ uncontrolled growth characteristic likely arises from the inability of their mitochondria to generate deuterium-depleted metabolic water as a consequence of a defunct tricarboxylic acid cycle [16].

Evidence on the anticancer potential of deuterium depletion in humans has been obtained in a prospective Phase 2 clinical trial involving prostate cancer patients [19] and retrospective studies on lung [20], breast cancer [21] and glioblastoma [22]. These studies, conducted on well-defined and homogeneous cancer populations in terms of cancer type and stage, revealed that when patients consumed DDW alongside conventional therapy, MST, depending on cancer type, increased three- to seven-fold compared to the historical control (five-fold in lung cancer, twofold in stage IV breast cancer, and threefold in glioblastoma). These findings underscore the therapeutic promise of deuterium depletion, positioning it as a groundbreaking mechanism and a promising direction for future cancer research and treatment development.

Cancer drug registration requires conducting prospective, randomized, placebo-controlled Phase 2 and Phase 3 clinical trials. However, the small sample sizes in clinical trials often limit the reliability of the results and conclusions, as they may not accurately reflect the true efficacy and safety of the investigated drug candidates. To address this limitation, collecting and analyzing real-world data is essential for a comprehensive understanding of a drug’s performance, ultimately enabling a more accurate evaluation of its efficacy in cancer treatment [23].

Given DDW’s demonstrated safety [19] and the absence of any dietary requirement for deuterium, drinking water with reduced deuterium content was introduced as a commercial food product in 1994 with approval from the Hungarian food safety organizations. This marked a significant step forward in making DDW accessible for broader use.

Patients consuming DDW voluntarily shared their experiences and outcomes before, during, and after DDW consumption, along with the results of follow-up oncology examinations confirming the effectiveness of conventional treatments. This valuable feedback was collected by HYD LLC and was processed in a comprehensive analysis across various tumor types, the published findings of which contributed to the growing body of evidence supporting DDW’s therapeutic potential.

In 32 years, reliable data have been collected on 2649 patients (the first enrolled patient started to consume DDW in October 1992 and the last one, in October 2024), forming an ample set of real-world data, the foundation of the present publication. In this study, we focused on survival data, the primary common descriptor of treatment outcome of any malignant disease.

Evaluation of the fate of cancer patients with and without DDW consumption requires a reliable base of comparison. The *2023 Country Cancer Profile: Hungary report* by the European Cancer Inequalities Registry was the foundation for conclusions that were subsequently compared with our data and findings. This comparison provided valuable context and insights into Hungary’s broader landscape of cancer care and outcomes [6].

## 2. Methods

### 2.1. Patients

The evaluated cancer population encompasses a diverse range of malignancies, including various organs of origin, distinct pathological backgrounds, different stages, and diverse treatment protocols. The final database included 2649 patients (Figure 1), comprising 43.67% males (1157) and 56.33% females (1492). The average age of the overall population was 55.63 years (median: 58 years). Among males, the average age was 56.3 years (median: 60 years), while for females, it was 55.1 years (median: 56 years). The average body weight across the population was 69.6 kg (median: 70), with males averaging 76.1 kg (median: 76 kg) and females averaging 64.7 kg (median: 64 kg).

### 2.2. DDW Production and Its Application for Human Use

Deuterium-depleted water was produced from ordinary water containing the natural concentration of deuterium (150 ppm, equivalent to 16.8 mmol/L). Fractional distillation was employed to reduce the deuterium (D) concentration to 50–25 ppm. This method is based on the differences in the physical properties of normal water (H_2_O) and heavy water (D_2_O)—specifically, that, in equilibrium with liquid water, steam at the boiling point of normal water contains approximately 2.5% less deuterium. The deuterium content of water was progressively reduced through repeated evaporation, which was performed on an industrial scale in distillation towers, and the distillate was measured. The instrument (Liquid-Water Isotope Analyzer-24d, manufactured by Los Gatos Research Inc., San Jose, CA, USA) uses off-axis integrated cavity output spectroscopy to measure the absolute abundance of D-containing water molecules via laser absorption. The deuterium concentration is given in ppm (with ±1 ppm accuracy as stated by the manufacturer).

To prepare the final products (that is, the drinking water Preventa 125, 105, 85, 65, 45, and 25), DDW was mixed with high-quality spring water to set the required D concentration and replenish minerals removed during distillation. For the latter purpose, a mineral stock solution was also used.

DDW is now commercially available under different brand names in several countries, including the United States of America, Russia, China, Japan, Romania, and Hungary. The production and sales of DDW have been steadily growing worldwide, reaching a yearly amount of approx. 600 tons by 2024.

To maximize the potential efficacy of DDW, patients are advised to consume it exclusively, at a rate of 1.5–2 L per day, depending on an average body weight of 60–80 kg. For individuals weighing over 80 kg, daily DDW intake should exceed 2 L. The quantity of DDW consumed should account for 75–80% of the total daily fluid intake, while the remaining 20–25% is expected to come from the water content in food. The optimal D concentration and the duration of consumption depend on various factors, including the type of cancer, its stage, and any ongoing conventional therapy. Generally, it is advisable to begin DDW consumption at either 105 ppm or 85 ppm, as these levels were found to be necessary for efficacy [21]. The primary concept behind DDW administration is to achieve a gradual, sustained reduction in the body’s deuterium levels over time, thereby challenging the metabolism of all cancer and healthy cells. To facilitate this, the DDW being consumed should be progressively replaced with another preparation with a D concentration 20 ppm lower than the previous level. The recommended duration for consuming DDW with a specific concentration is typically 2–3 months. However, this can vary depending on factors such as the type of conventional therapy, cancer localization, pathology, and the individual’s response to conventional treatment and DDW consumption [24].

### 2.3. Integration of DDW into Conventional Therapies

The use of DDW is not classified as a treatment. Our study was designed to collect data from cancer patients undergoing conventional therapies such as chemotherapy, targeted drug therapy, radiotherapy, hormone therapy, or surgery while consuming DDW. The oncologists’ treatment protocols remained untouched and no extra patient examinations were conducted. Instead, the information generated during the participating patients’ conventional therapies was provided voluntarily to HYD LLC, which collected and processed it and gave advice on DDW consumption based on the patients’ medical history and the results of follow-up examinations.

A set of detailed protocols was developed for patients suffering from various malignancies in different stages based on insights from over 30 years of follow-up studies. These are specified in the book *Deuterium Depletion—A New Way in Curing Cancer and Preserving Health* [24]. Studies, including a landmark article on lung cancer, further support the value/usefulness of these protocols, underscoring the benefits of DDW consumption [20].

### 2.4. Evaluation of Provided Data of the DDW-Consuming Cancer Patients

This population-based observational retrospective study evaluated the data accumulated over 32 years (the first patient was enrolled on 29 October 1992 and the last one on 18 October 2024). All patients enrolled in the study were diagnosed with cancer, and while receiving conventional therapy, they also consumed DDW in parallel, as stated above. As previously explained, DDW consumption was the only difference from the overall Hungarian cancer population used as the comparison base. The evaluated patient population was highly heterogeneous regarding tumor types, stages, conventional therapies received, and the duration of DDW consumption. Given this variability, the evaluation of survival data was chosen as the study’s primary endpoint; that is, the survival times of the DDW-consuming patients were compared to the historical control described in the next section to draw meaningful conclusions.

The following information was available for the participants: pseudonymization code, gender, age at the start of DDW consumption, body weight, primary cancer localization, date of diagnosis, date of DDW initiation, staging at the beginning of DDW consumption (remission or active tumor presence), date DDW consumption ended, date of the last recorded information, and the patient’s status (alive or deceased) at that time. The medical history of patients with DDW consumption was divided, according to the above-mentioned data, into three primary phases, with two additional phases calculated from these (Figure 2) as follows:Time from Diagnosis to the Start of DDW Consumption: The period between the ini-tial diagnosis and the beginning of DDW treatment.Duration of DDW Consumption: The total time the patient consumed DDW.Follow-Up Time After Stopping DDW Consumption: The duration from the cessation of DDW consumption to the end of the follow-up period.Total Time from Diagnosis to the End of Follow-Up: The combined time spanning from diagnosis to the conclusion of follow-up.Time from the Start of DDW Consumption to the End of Follow-Up: The interval from the initiation of DDW treatment to the end of the follow-up period.

**Figure 2 biomedicines-13-00876-f002:**
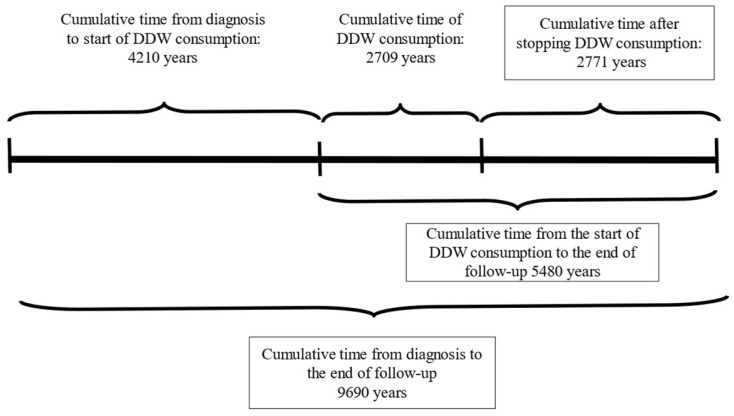
Cumulative Duration Times of 2649 Patients from the Date of Diagnosis to the End of Follow-up.

### 2.5. The Base of Comparison: Cancer Mortality and Estimated Median Survival Time in the US, Europe and Hungary

The general population’s MST from cancer is approximately 2.7–3 years in the United States and Europe, estimated from the number of new cancer cases and the cancer-caused deaths every year. In Hungary, cancer-related mortality was 33% higher than the EU average in 2019 [3]. Cancers such as colorectal, lung, pancreatic, and stomach account for 52% of all cancer-related deaths in Hungary. These cancers, except colorectal, typically have an MST of just over one year or less [6]. Consequently, Hungary’s overall cancer MST is estimated to be lower than the European average, around 2.4 years.

Accordingly, the 2.4-year MST is considered as a historical control for this analysis, against which data from the DDW-consuming population will be compared.

### 2.6. Endpoints and Assessment

Given the heterogeneity of the enrolled population, survival was selected as the primary endpoint for assessing the efficacy of deuterium depletion. Earlier studies conducted on small, homogeneous cancer subpopulations demonstrated several key findings: prolonged DDW consumption correlated with extended survival; lower deuterium concentrations of DDW were associated with higher response rates, particularly in breast cancer patients [21]; and different tumor types exhibited varying degrees of sensitivity to deuterium depletion, as observed in lung cancer [20,25] and glioblastoma multiforme [22]. These studies, importantly, also proved the validity of the available data and the analysis method. The current study expanded the evaluation to include the entire population, but to ensure robustness and validation, analyses also focused on specific subgroups. This approach is expected to provide a comprehensive understanding of the relationship between deuterium depletion and cancer progression/regression and to evaluate the potential of integrating deuterium depletion into conventional therapy.

### 2.7. Study Oversight

This study is a population-based observational retrospective analysis based on data collected over 32 years by HYD LLC. Patients consuming deuterium-depleted drinking water (marketed under the brand name Preventa^®^) already available on the market voluntarily shared their experiences and the results of their follow-up examinations. All patients received standard oncological treatments according to established protocols, and no intervention was made beyond monitoring their DDW consumption.

The data evaluation process involved collaboration with experienced statisticians and leading oncologists to ensure a thorough and validated analysis and interpretation of the results.

### 2.8. Statistical Analysis

The primary endpoint of the study was survival. Kaplan–Meier survival estimates and log-rank tests were used to compare groups. The statistical analysis compared the MST value of the 2649 patients with the historical control data, calculated at 2.4 years. A detailed analysis of the underlying factors behind the given MST value was conducted to minimize the likelihood of potential errors. We examined the factors influencing MST and analyzed and investigated the relationships and interactions among multiple factors. The average time from diagnosis to the commencement of DDW consumption was 19 months. To accurately assess the efficacy of DDW consumption, the MST was calculated from both the date of diagnosis and the initiation of DDW consumption, as the patients already exhibited prolonged survival from the time of diagnosis before starting DDW. All statistical computations were performed by using the software SPSS v25. This study was performed retrospectively, and all statistical results were declared significant with *p* < 0.05. For correlation analysis, the Pearson method was used. The calculations were performed by Adware Research Ltd. (Balatonfüred, Hungary).

## 3. Results

### 3.1. Characteristics of the Patient Population

The 2649 patients were divided into two primary groups in terms of staging: 2393 patients (90.4%) had active tumors at the start of DDW consumption (Tumor Group, TG), while 256 patients (9.6%) were in complete remission (Remission Group, RG). The TG included 1083 males (45.2%) and 1310 females (54.8%), while the RG comprised 75 males (29.2%) and 181 females (70.8%). Figure 1 illustrates that, over a cumulative follow-up period of 111,965 months (9690 years) to the end of the study, 609 patients (22.9% of the total evaluated population of 2649) died while 2040 patients were still alive. Among the surviving patients, 88.3% belonged to the TG and 11.7% to the RG. Among the deceased, 306 (50.2%) patients were male and 303 (49.8%) were female, representing 26.5% of the male population and 20.3% of the female population.

Notably, 591 (97%) of the deceased individuals (299 males and 292 females) were from the TG, while 18 individuals (7 males and 11 females), accounting for 3% of the deceased population, belonged to the RG.

The patients were divided into 12 groups based on their primary tumor types (see Table 1), and the percentage distribution of the types was compared to the distribution of the same tumor types in the Hungarian cancer population.

The data reveal strong parallels between the distribution of cancer types in the DDW-consuming population and the overall Hungarian cancer population. The four major cancer categories—digestive, breast, lung, and urological cancers—accounted for 66.7% of the DDW-consuming population, closely aligning with the 68% observed in the Hungarian cancer population. Among the smaller cancer groups, brain tumors were notably overrepresented in the DDW-consuming cohort, while gynecological tumors were underrepresented compared to the national distribution.

This substantial similarity between the two groups validates comparing data from the DDW-consuming population to the calculated median survival time for the Hungarian cancer population.

### 3.2. Survival Outcomes

#### 3.2.1. MST Calculation for the Entire Study Population

In the first step of the analysis, the entire patient population was examined without imposing any restrictions on the duration of DDW consumption or the disease stage at initiation. Table 2 summarizes the cumulative follow-up periods for all 2649 patients, from the date of diagnosis to the end of the follow-up.

As shown in Table 2, half of the patients began consuming DDW not more than 3.8 months after their tumor diagnosis, with an average time of 19.0 months. Additionally, 50% of the patients consumed DDW for more than half a year (7.3 months), while the average duration of DDW consumption exceeded one year (12.2 months). Furthermore, half of the patients were followed for nearly two years (22.7 months). Figure 3 shows the Kaplan–Meier curve calculated from the date of diagnosis (A) and Pearson’s correlation between the length of DDW consumption and survival time (B). The MST from diagnosis was 12.4 years (149.0 months; 95% CI: 118.6–179.3). In Figure 4, analogous data, calculated from the start of DDW consumption, are presented. The MST from the start of DDW treatment was 7.6 years (92.2 months; 95% CI: 71.8–112.5). The correlation between the length of DDW treatment and survival times was statistically significant, calculated both from the start of diagnosis (Pearson’s coefficient: r = 0.476, *p* < 0.001) and from the start of DDW treatment (r = 0.635, *p* < 0.001). The correlation between survival times calculated from the start of diagnosis and those calculated from the start of DDW treatment was also statistically significant (r = 0.717, *p* < 0.001).

#### 3.2.2. MST Calculation for the Tumor Group

As shown in Figure 1, two groups of patients were created regarding the staging, namely, 256 patients out of 2649 started to consume DDW while in complete remission (remission group, RG), and 2393 had tumors at the time of commencing DDW consumption (Tumor Group, TG). The good prognosis of the 256 patients in the RG (only 3% of them died during the follow-up period) strongly increased the MST of the entire study population. Hence, the next step was to evaluate the MST of the 2393 patients in the TG. Given the dissimilar efficacy of therapies in tumors of various organs (localization), it was essential to compare the distribution of tumor types in the TG and RG with that in the whole population before the MST for the TG was calculated.

The numbers in Table 3 show that removing the RG (less than 10% of the cases) from the entire population did not modify the composition of the TG much compared to the whole study population. Hence, the median survival time (MST) of the 2393 patients in the TG remained comparable to the historical controls. The composition of the RG, in contrast, differed considerably from that of the whole patient population, resulting from the mentioned varying efficacy of conventional therapies. Breast cancer was strongly overrepresented (34.8% in the RG vs. 17.9% in the general population), reflecting the high success rates of conventional treatments for this cancer type. Conversely, lung cancer is notably underrepresented (4.3% vs. 15.8%), consistent with the fact that only 10% of lung cancer cases are operable. The differences regarding the other cancer types are minor.

The TG is considered a homogeneous population, as all patients had tumors at the time they began consuming DDW. The MST in the whole TG was 10.9 years when calculated from diagnosis and 5.8 years from the start of DDW consumption—ca. 20% shorter than the corresponding data of the whole study population. Even so, the MST of the 2393 TG patients showed a significant increase—4.5-fold and 2.4-fold—compared to the Hungarian cancer population, which had an MST of 2.4 years.

#### 3.2.3. MST Calculation Considering the Mortality and the Duration of DDW Consumption

As shown above, the inclusion of the RG in the MST calculation increased the overall MST. It is also evident that mortality cases strongly influence the MST. During the cumulative follow-up period of 9690 years, 609 patients died (Table 2), with 97% of these cases belonging to the TG. It is of note that the average time between diagnosis and the initiation of DDW consumption was similar for TG patients who remained alive and those who had died (19.9 vs. 19.1 months), suggesting that the patient’s general stage at the start of DDW consumption was the primary determinant of their outcomes. To prove that, the number of deaths was counted month by month in the first two years, separately from the date of diagnosis and the start of DDW consumption.

As shown in Table 4, 228 cancer patients passed away within the first eight months from the start of DDW consumption. Analyzing mortality cases based on both the date of diagnosis and the initiation of DDW consumption revealed that 80 out of these 228 deaths occurred within the first eight months from diagnosis, while the remaining 148 patients had a more extended medical history—exceeding eight months—before starting DDW consumption.

Notably, for the 80 patients with short survival, the average time between diagnosis and the start of DDW consumption was just one month, indicating that the mortality of the 80 cases was likely due to a poor prognosis at the time DDW consumption was initiated and not to any delay in initiation. For the other 148 patients, the average period was 27.9 months. This 27.9-month duration is eight months longer than the average calculated for the entire population.

Another critical factor to consider is the duration of DDW consumption. Table 4 provides data on the number of fatalities, the average time elapsed between diagnosis and the initiation of DDW, and the length of DDW consumption of the 228 patients.

The data demonstrate that from the third month onward, patients who passed away often had a long medical history before starting to consume DDW. Additionally, for those who died within the first two to three months, the duration of DDW consumption was remarkably short (0.5–2 months). These phenomena point to limiting factors, such as the minimum time required for DDW to exert its effect and whether the patient’s condition is still (or is no longer) reversible or stoppable. To determine the boundaries by means of MST calculation, we again took the whole TG and excluded from the evaluation, month by month, those patients who passed away within the first eight months of DDW use (and these data are presented in Table 4). Table 5 shows the number of patients evaluated and the corresponding MSTs.

It is important to note that the 228 patients who died within eight months after DDW consumption represent 37% of the 609 patients who died during the entire follow-up period of 9690 years, indicating that these patients were in an advanced stage of the disease at the start of DDW consumption, or more probably, even at the time of diagnosis. The cumulative follow-up time of the 228 patients was 444 years, representing only 4.5% of the cumulative follow-up time of the entire cancer population, and 79% (351 years) of the 444 years elapsed between the diagnosis and the start of DDW consumption, confirming that these patients’ disease dragged on for a long time, possibly years, before they decided to start with DDW. When excluding patients who passed away within the first three months of DDW consumption, the MST from the beginning of DDW consumption increased from 70 months to 76 months. Excluding early deaths does, of course, raise the MST—but these data (and experiences of application) suggest that consuming DDW for a sufficiently long time contributes to extended survival, even in patients with late-stage conditions. To explore this latter possibility, the relationship between the length of DDW consumption and MST was investigated.

A minimum volume of DDW must obviously be consumed to reduce the internal deuterium concentration sufficiently to induce a tumor response. To see the effect of the duration (and hence, overall volume) of DDW consumption, the MST of the whole tumor group (described above) was taken and re-calculated with the step-by-step exclusion of patients consuming DDW for less than 31, 61, 91, 121, 151, and 181 days (the patient cohort was highly heterogeneous regarding the duration of DDW consumption, with some individuals consuming it for only a few days or months while others continued consuming it for several months or even years). Table 6 summarizes the patients evaluated and the MST calculated from the diagnosis and start of DDW consumption.

The data indicate a four-month increase in MST when calculated from the start of DDW consumption for the population consuming DDW for more than 90 days and an additional four-month increase for those consuming DDW for over 120 days (that is, the rise of MST was in both cases longer than the increase in the duration of DDW use). Table 6 supports the earlier described correlation, namely, that prolonged consumption of DDW is associated with extended survival. Based on these findings, it can be concluded that DDW requires a consumption period of at least 90, but better 120, days to exert its beneficial effects.

#### 3.2.4. MST Among Patients Who Met the Above Criteria of Life Expectancy and Length of DDW Consumption

The data presented in the previous sections indicated that consuming DDW during the late stage of cancer provided limited benefits and that a minimum length of DDW consumption was necessary to impact survival. Namely, a successful DDW application requires more than 90–120 days of DDW consumption and a corresponding life expectancy of 3–4 months. The following assessment was focused on the population that met both prerequisites. The calculation was performed with two exclusion criteria as follows:

When patients with <91 days survival and <91 days of DDW consumption were excluded, 2035 patients remained (group: 91SURV91DDW), and the MST from diagnosis was 10.1 years (121.6 months, 95% CI: 100.0–143.3). From the initiation of DDW treatment, it was 6.2 years (74.5 months, 95% CI: 61.2–87.8).

When the exclusion was extended to patients with <121 days survival and <121 days DDW consumption, 1758 patients remained (group 121SURV121DDW). The MST from diagnosis was 11.0 years (132.3 months, 95% CI: 107.2–157.3), and from the initiation of DDW treatment, 6.5 years (78.1 months, 95% CI: 62.6–93.6).

In both cases, as with the entire population (Figure 3 and Figure 4), the correlation between the duration of DDW treatment and survival times—whether calculated from diagnosis or the start of DDW treatment—was statistically significant (r = 0.460, r = 0.456, *p* < 0.001; r = 0.643, r = 0.634, *p* < 0.001). The correlation between survival times from diagnosis and the start of DDW treatment was also statistically significant (r = 0.683, *p* < 0.001).

These findings show that among those cancer patients who survived and consumed DDW long enough for deuterium depletion to exert its effect, DDW consumption extended the historical control survival time of 2.4 years to an MST of 10.1–11.0 years from diagnosis and 6.2–6.5 years from the initiation of DDW treatment. These findings, along with the data in Table 6, demonstrate that even minor variations in the duration of DDW consumption (and similarly, the time between diagnosis and the initiation of DDW, see Table 4) intake can lead to multiplied changes in the MST.

#### 3.2.5. DDW Consumption Delays the Progression of the Disease and Prevents Relapses Among the RG

The cumulative follow-up time of 256 patients who began consuming DDW while in complete remission amounted to 14,289 months (1190 years). This group included all 12 cancer types, as detailed in Table 7. By the data collection cut-off date of 12 November 2024, 18 patients had passed away. Table 7 presents the distribution of these losses across different types of cancer, offering insights into mortality patterns within this cohort.

The extremely low mortality rate (7%) in the RG made calculating the MST from the time of diagnosis impossible. However, when calculated from the start of DDW consumption, the MST for RG patients was 23.2 years.

It is of interest that the time gap from diagnosis to start of DDW consumption was, on average, half as long in the RG (10.5 ± 23.7 months, median: 2.8 months) as in the TG (20.0 ± 38.9 months, median: 4.0 months). This suggests another critical factor, the time that elapsed between the diagnosis and the start of DDW consumption. The MST of those patients in the entire cancer population who started to consume DDW within 3, 6, or 9 months after diagnosis was stable and high, 11.1–11.3 years. However, there was an almost two-year drop in MST (9.6 years) involving those who started consuming DDW over 12 months after their diagnosis.

This finding underscores the critical role of early cancer detection combined with timely and appropriate therapy that can lead to a complete remission. In the RG, all patients began consuming DDW within one year of their diagnosis, still free of disease progression.

#### 3.2.6. Evaluation of MST by Unifying the Patient Groups Where DDW Application Was Found to Be Efficient

After applying the above defined criteria of life expectancy and length of DDW consumption and including the 256 patients in the RG group, 2291 and 2014 patients were analyzed.

When the time limit of DDW use was set to <91 days, the median survival time of the 2291 patients (91SURV91DDW + 256) from diagnosis was 12.2 years (146.9 months; 95% CI: 120.0–173.9), and from the start of DDW treatment, 8 years (96.4 months; 95% CI: 76.5–116.3). The correlation between the length of DDW treatment and survival times calculated both from diagnosis (r = 0.466, *p* < 0.001) and from the start of DDW treatment (r = 0.616, *p* < 0.001) was statistically significant.

Setting the time limit to <121 days, 2014 patients (121SURV121DDW + 256) were evaluated. Their median survival time from diagnosis was 13 years (156.2 months; 95% CI: 120.8–185.5), and from the start of DDW treatment, 8.7 years (105.0 months; 95% CI: 82.8–127.1). The correlation between the length of DDW treatment and survival times was significant, calculated both from the start of diagnosis (r = 0.462, *p* < 0.001) and the beginning of DDW treatment (r = 0.604, *p* < 0.001). The correlation between survival times calculated from the start of diagnosis and the start of DDW treatment was also statistically significant (r = 0.730, *p* < 0.001). These data show again that, for the anticancer effect of DDW to take place, the application should not be shorter than 90 days.

To check if the above selection caused any noteworthy distortion in the data, the distribution of different cancer types, after excluding patients with limited survival times and insufficient DDW consumption, was compared to the original distribution of cancer types.

As seen in Table 8, patients with digestive cancers are underrepresented in both selected groups, while breast and lung cancer cases became slightly overrepresented compared to the composition of the total study population. Aside from that, however, the overall distribution of cancer types within the population remained largely unchanged after excluding patients with early death and short-term DDW consumption.

#### 3.2.7. Correlation of DDW Consumption and Survival in Patients Who Were Followed up Until Death

At the level of individual patients involved in the study, the end of follow-up was when no more data were received, not necessarily their death. It was supposed, however, that the DDW consumption and survival data of those who were followed up until death could demonstrate the survival prolongation by deuterium depletion more clearly. In Figure 5, a dense cluster of data points delineate the straight line “y = x” showing the fate of patients who used DDW up to death. Other data points, scattered upwards, belong to patients who consumed DDW for a given period but lived on without further deuterium depletion. Such cases (observable also in Figure 3B and Figure 4B) may provide the best proof of the beneficial effect of deuterium depletion on the survival of cancer patients.

## 4. Discussion

There is no doubt that significant advances have been made in cancer therapy over the past decades. However, prognoses show that current treatments have not led—and are unlikely to lead—to a true breakthrough in oncology [7]. The greatest paradox of the prevailing approach is that, despite recognizing the immense complexity of genetic and biochemical processes of the cells, we seek solutions by targeting specific genetic mutations or using single molecules to selectively inhibit or influence isolated processes or, more precisely, the functional proteins responsible for them.

More than forty years ago, Albert Szent-Györgyi raised the question of whether harmonic regulation of the rapid and complex biochemical and genetic processes of a cell is (or can be) realized at the molecular level. He assumed that large protein molecules (the typical main targets of drug development) were incapable of regulating these processes, and stated that a sub-molecular regulatory system, based on electrons—very lightweight and mobile elementary particles—fulfills this role [26]. The data obtained with DDW [7,8,9,10] confirmed that sub-molecular changes, namely, an increase or decrease of D/H ratio, significantly impact the processes within the cells.

This cellular effect of altering the D/H ratio by deuterium depletion resulted at a systemic level in a severalfold increase in the median survival time (MST) of cancer patients consuming DDW in the studies referred to above [19,20,21,22] compared to historical controls. These findings underscore the importance of identifying key parameters for fully harnessing the potential of deuterium depletion when integrating it into conventional cancer therapy.

The results of the present study, based on real-world data of 2649 patients consuming DDW, showed the possible significant impact of integrating deuterium depletion into conventional oncotherapy, and highlighted both its potential and limitations. The MST for the patients involved increased to 12.4 years from the time of diagnosis and to 7.6 years from the start of DDW consumption, representing a five- and three-fold increase compared to the 2.4-year MST of the overall Hungarian cancer population. Among the patients with detectable tumors (the TG group), their MST was 5.8 years from the initiation of DDW consumption. In the RG group (patients in remission when starting deuterium depletion), their MST from the start of DDW consumption was 23.2 years.

To find the optimal conditions for integrating DDW into conventional therapy, subgroups within the 2649 cases were defined based on time parameters (see Table 4, Table 5, Table 6 and Table 8), and their MSTs were analyzed. It was concluded that, to maximize the efficacy of DDW, a life expectancy of not less than 3–4 months at the start of DDW consumption is necessary, the length of DDW consumption should be longer than 90–120 days, and DDW consumption should start not later than 9 months after diagnosis. The evidence for these requirements, the MST data for all the subgroups calculated from the start of DDW consumption, are summarized in Table 9.

The data showed that excluding patients with a short application period (who died within 90–120 days after starting DDW or who consumed DDW for less than 90–120 days) increased the MST from 5.8 years to 6.1–6.5 years. When both criteria were applied, the MST for patients who lived longer than 90–120 days and consumed DDW for more than 90–120 days went up to 8.0–8.7 years.

However, the most critical factor, highlighted by the numbers in Table 9, was the time gap between diagnosis and initiation of DDW consumption. When DDW consumption started within 9 months after diagnosis, the MST rose to 11.3 years (95% CI: 8.4–14.1), but it dropped to 9.6 years (95% CI: 7.2–11.8) starting over a year after diagnosis. When the calculation was restricted to patients who lived longer than 120 days after starting DDW and consumed it for more than 120 days, only a minimal additional increase in MST (to 11.6 years) was observed.

This population, who began to use DDW within 9 months after diagnosis, achieved a nearly 5-fold increase in MST compared to the average Hungarian cancer population, emphasizing the importance of timely and sustained DDW consumption in improving survival outcomes.

With an MST of 11.6 years, Hungary’s annual cancer death toll could drop from 33,000 to approximately 5000–7000, potentially saving 26,000–28,000 lives each year. In Europe, the cancer death toll could be reduced by 825,000 to 880,000 lives through the integration of deuterium depletion into conventional therapy. This represents a dramatic improvement for the cancer population, underscoring the potential life-saving impact of integrating DDW consumption into therapy.

Deuterium depletion constitutes a paradigm-shifting approach by abandoning the focus on single targets within the cell regulation. Gene expression studies indeed demonstrated that altering the deuterium concentration had a profound impact on entire cells or organisms.

In humans, 200–300 genes regulate the cell cycle directly, while many additional genes indirectly engage in related signaling pathways and repair mechanisms. Two previous studies [25,27] demonstrated that the expression of specific genes associated with tumor development, such as c-Myc and K-Ras, H-Ras, Bcl2, p53, was inhibited in carci-nogen-treated mice consuming DDW. This suggests a potential role of the intracellular deuterium level in modulating gene expression linked to tumorigenesis, offering promising insights for future research and therapeutic applications.

An elevated D/H ratio seems thus essential for triggering the expression of specific genes associated with tumor development. A recent study [28] utilizing nanostring technology investigated the expression of 236 cancer-related genes and 536 kinase genes in deuterium-depleted (40 and 80 ppm D), deuterium-enriched (300 ppm), and regular (150 ppm) media. From the total, 124 cancer-related genes and 135 kinase genes (those with expression changes exceeding 30% and a copy number above 30) were evaluated.

Only seven genes exhibited altered expression (one upregulated and six downregulated) in deuterium-depleted media, but 97.3% of the evaluated genes were upregulated after deuterium enrichment. These findings align with the above-mentioned in vivo mouse studies, demonstrating that DDW keeps the D concentration at a low level and inhibits its rise to the threshold necessary to trigger the expression of genes induced by carcinogens. Research confirmed that consuming DDW prevents the increase in the D/H ratio, thereby blocking the activation of the entire set of genes involved in cell cycle regulation. In harmony with these results, a cell cycle analysis revealed that DDW caused cell cycle arrest in the G1/S transition, reduced the number of cells in the S phase, and significantly increased the population of cells in the G1 phase [29].

DDW reduces the body’s deuterium concentration [30], effectively mimicking the role of mitochondria in producing deuterium-depleted metabolic water. Studies suggest that lowering deuterium levels impacts cellular metabolism and generates free radicals [31], demanding rapid and efficient adaptive responses. Our findings suggest that the differential adaptive responses of healthy and cancer cells to free radicals render deuterium-depleted water selectively effective against cancer cells. Healthy cells with fully functional mitochondria and metabolism can successfully manage this metabolic challenge. In contrast, cancer cells, which typically lack such mitochondrial functionality, fail to adapt, leading to apoptosis, necrosis, and observable tumor regression [9,10]. This is further supported by our observation that DDW consumption enhances the efficacy of radiotherapy, a treatment modality that relies on the generation of free radicals [22].

Safety is a primary concern in the case of any novel therapeutic method to be applied in humans. As for deuterium depletion, the consumption of DDW with 25–125 ppm deuterium content has proven to be safe, with no unexpected toxic or harmful effects observed [19,32,33]. In contrast, some beneficial effects beyond tumor growth inhibition were observed. In a Phase 2 clinical trial involving 30 patients with pre or manifested diabetes, significant increases in blood cell counts within the normal range and a reduction in fasting blood glucose levels were observed [32]. These findings were consistent with observations in cancer patients undergoing chemotherapy and consuming DDW, where significant blood count deterioration was either absent or delayed. A general improvement in the patients’ physical strength and well-being, associated with DDW consumption, was also repeatedly found. The underlying mechanism is likely the one described in a test with top athletes, where a 44-day regimen of 105 ppm DDW consumption resulted in a delayed increase in lactic acid levels and a reduced anion gap, indicating more efficient mitochondrial function [33].

On the other hand, tumor necrosis caused by deuterium depletion was not without some kind of side effects. The most characteristic changes were weakness; drowsiness; increased temperature and fever spikes; intermittently increasing pain; swelling and softening of the tumor-affected area; minor bleeding in the bladder, stomach, or rectum; brick dust urine; and transient coughing in lung cancer [24]. All these side effects were tolerable by the patients and were transient.

In several cases, patients involved in the present study used various supplementary or alternative treatments in parallel with conventional oncotherapy and DDW consumption, and the data suggested a possible negative interaction between deuterium depletion and certain supplementary methods. After a thorough evaluation, several factors were identified that significantly diminished the effectiveness of DDW or rendered it entirely ineffective. These included high doses of antioxidants, such as vitamins A, C, E, and selenium; consumption of Coenzyme Q10 supplements; iron supplementation; intense and prolonged physical exertion; as well as the use of hot tubs and saunas. Understanding these influences can provide critical guidance for optimizing the efficacy of DDW application and highlights the importance of managing patient lifestyles and supplementary treatments during deuterium depletion.

The groundbreaking findings in this study underscore the pivotal role of the D/H ratio in regulating and coordinating millions of molecular processes, including biochemical and genetic functions involved in carcinogenesis. At the eukaryotic level, a sub-molecular regulatory system (SMRS) governs the intricate complexity of life. Utilization of the capacities of DDW enables an intervention in this fundamental regulatory system, opening new avenues for therapeutic applications and a deeper understanding of cellular processes.

## Figures and Tables

**Figure 1 biomedicines-13-00876-f001:**
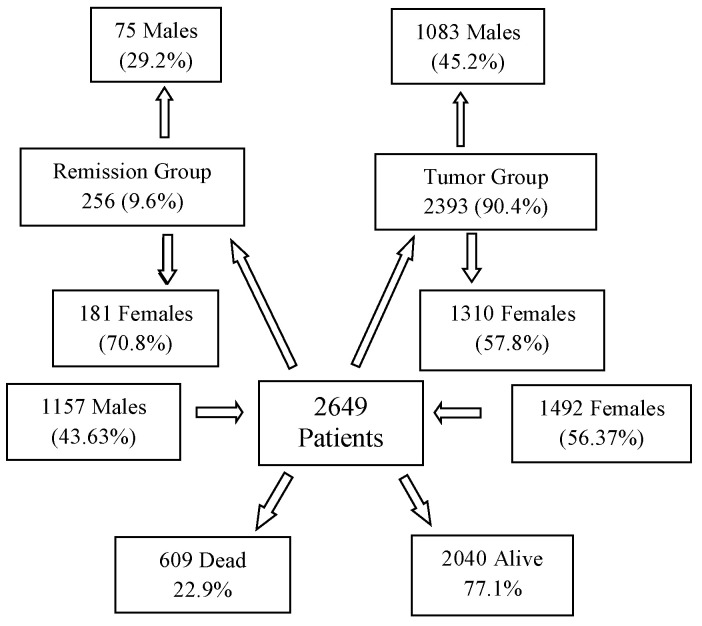
Subgroups of 2649 Patients Based on Gender and Staging at the Start and the End of DDW Consumption.

**Figure 3 biomedicines-13-00876-f003:**
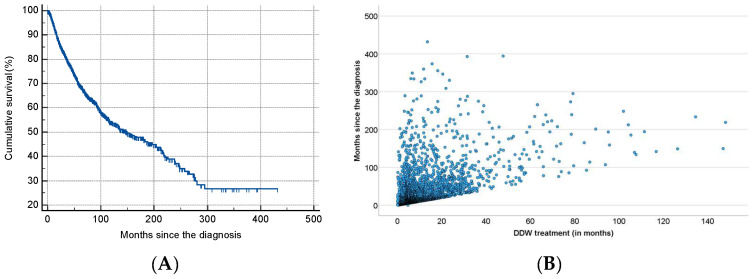
Kaplan–Meier Curve of 2649 Patients Calculated from the Date of Diagnosis (**A**) and Pearson’s Correlation Between the Length of DDW Consumption and Survival (**B**).

**Figure 4 biomedicines-13-00876-f004:**
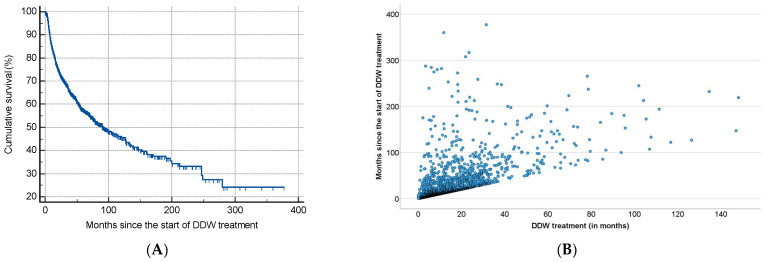
Kaplan–Meier Curve of 2649 Patients Calculated from the Date of Starting DDW Consumption (**A**) and Pearson’s Correlation Between the Length of DDW Consumption and Survival (**B**).

**Figure 5 biomedicines-13-00876-f005:**
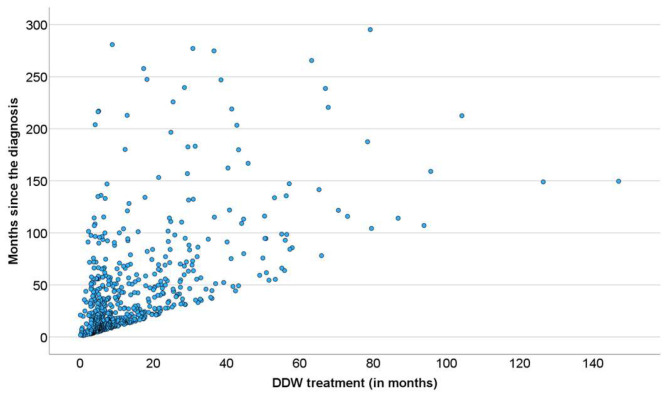
Pearson’s Correlation Between the Length of DDW Consumption and Survival in the 609 Patients Who Were Followed up until Death.

**Table 1 biomedicines-13-00876-t001:** Percent Distribution of Major Cancer Types in the Follow-Up Population (N = 2649).

Primary Localization of the Tumor	Number of Enrolled Patients	Percentage of the Cancer Types in the DDW-Consuming Population	Percentage of the Cancer Types in the Hungarian Cancer Population
Digestive	529	20.0	23
Breast	475	17.9	20
Lung	419	15.8	15
Urological	345	13.0	10
Hematopoietic	212	8.0	5
Brain	189	7.1	2
Gynecological	195	7.4	12
Head and neck	106	4.0	3
Bone and soft parts	73	2.8	1
Skin	63	2.4	5
Juvenile	16	0.6	1
Other	27	1.0	3

**Table 2 biomedicines-13-00876-t002:** Summary of the Cumulative Durations for Various Phases of the Follow-up Periods, Along with the Calculated Median and Average Durations per Patient.

	Years	Median (Months)	Average and SD (Months)
Cumulative time elapsed from diagnosis to the start of DDW consumption	4210	3.8	19.0 ± 37.8
Cumulative duration of DDW consumption	2709	7.3	12.2 ± 15.4
Cumulative duration of follow-up after stopping DDW consumption	2771	0	6.6 ± 25.6
Cumulative time elapsed from diagnosis to the end of follow-up	9690	22.7	43.8 ± 54.2
Cumulative time elapsed from the start of DDW consumption to the end of follow-up	5480	10.3	24.8 ± 39.7

**Table 3 biomedicines-13-00876-t003:** Percent Distribution of Major Cancer Types in the Tumor Group and Remission Group.

Primary Localization of Cancer	Patients in TG		Patients in RG		Patients in the Whole Study Population, Percentage
	Number	Percentage	Number	Percentage	
Digestive	482	20.1	47	18.4	20.0
Breast	386	16.1	89	34.8	17.9
Lung	408	17	11	4.3	15.8
Urological	318	13.3	27	10.5	13.0
Hematopoietic	222	8.4	10	3.9	8.0
Brain	172	7.2	17	6.6	7.1
Gynecological	170	7.1	25	9.8	7.4
Head and neck	94	3.9	12	4.7	4.0
Bone and soft parts	70	2.9	3	1.2	2.8
Skin	53	2.2	10	3.9	2.4
Juvenile	13	0.5	3	1.2	0.6
Other	25	1.0	2	0.8	1.0

**Table 4 biomedicines-13-00876-t004:** Average timing of DDW consumption of the 228 patients in the TG who died during the follow-up.

Months Between Start of DDW and Death	Number of Death Cases	Average Time Between Diagnosis and Start of DDW (Months)	Average Length of DDW Consumption (Months)
1	6	3.6	0.5
2	10	5.7	1.3
3	12	19.9	2.0
4	44	20.0	3.3
5	39	17.4	4.3
6	46	17.9	5.7
7	36	20.4	6.2
8	35	20.6	7.0

**Table 5 biomedicines-13-00876-t005:** Number of Patients Evaluated After Excluding Those Who Died Within the First Eight Months of DDW Consumption, and the Calculated MSTs From the Date of Diagnosis and the Start of DDW Consumption.

Months Between Start of DDW and Deaths	Number of Patients Excluded	Number of Patients Evaluated	MST from the Diagnosis (Months)	MST from the DDW Start (Months)
0	0	2393	131	70
1	6	2387	131	70
2	10	2377	132	71
3	12	2365	134	74
4	44	2321	147	76
5	39	2282	153	78
6	46	2236	180	85
7	36	2200	187	90
8	35	2165	203	96

**Table 6 biomedicines-13-00876-t006:** Number of Patients Evaluated After Excluding Those Who Consumed DDW for a Limited Time and the Calculated MST from the Date of Diagnosis and the Start of DDW Consumption.

Length of DDW Consumption	Number of Patients	MST from Diagnosis (Months)	MST from DDW Consumption (Months)
Longer than 1 day	2393	131	70
Longer than 30 days	2287	128	70
Longer than 60 days	2168	121	71
Longer than 90 days	2035	121	74
Longer than 120 days	1758	132	78
Longer than 150 days	1563	134	82
Longer than 180 days	1378	146	89

**Table 7 biomedicines-13-00876-t007:** Distribution of Cancer Types in Remission Group and Number of Death Cases.

Primary Localization of Cancer	Number of Patients	Percentage	Number of Deaths
Digestive	47	18.4	7
Breast	89	34.8	2
Lung	11	4.3	0
Urological	27	10.5	2
Hematopoietic	10	3.9	2
Brain	17	6.6	2
Gynecological	25	9.8	1
Head and neck	12	4.7	1
Bone and soft parts	3	1.2	0
Skin	10	3.9	1
Juvenile	3	1.2	0
Other	2	0.8	0

**Table 8 biomedicines-13-00876-t008:** Distribution of Different Cancer Types, after Excluding Patients with Limited Survival Times and Insufficient DDW Consumption.

Primary Localization of Cancer	91SURV91DDW + 256		121SURV121DDW + 256		Percentage in the DDW-Consuming Population	Percentage in the Hungarian Cancer Population
	Number	Percentage	Number	Percentage		
Digestive	428	18.7	370	18.4	20.0	23
Breast	418	18.2	388	19.3	17.9	20
Lung	368	16.1	322	16.0	15.8	15
Urological	305	13.3	261	13.0	13.0	10
Hematopoietic	191	8.3	163	8.1	8.0	5
Brain	162	7.1	142	7.1	7.1	2
Gynecological	171	7.5	151	7.5	7.4	12
Head and neck	94	4.1	84	4.2	4.0	3
Bone and soft parts	65	2.8	57	2.8	2.8	1
Skin	51	2.2	44	2.2	2.4	5
Juvenile	16	0.7	14	0.7	0.6	1
Other	22	1.0	18	0.9	1.0	3

**Table 9 biomedicines-13-00876-t009:** MST Data for the Whole Study Population and Subgroups Defined with the Parameters Explained in the Footnote.

Name of the Group	Number of Patients	MST from the Start of DDW Consumption (Years)
The Entire Group of Patients (EG = TG + RG)	2649	7.6
TG	2393	5.8
RG	256	23.2
TGSURV91	2365	6.1
TGSURV121	2322	6.3
TGDDW91	2035	6.1
TGDDW121	1758	6.5
EGSURV91DDW91	2291	8.0
EGSURV121DDW121	2014	8.7
EGDDW within 9 months	1651	11.3
EGSURV91DDW91DDW within 9 months	1414	11.3
EGSURV121DDW121DDW within 9 months	1243	11.6
Hungarian historical control	70,000	2.4

## Data Availability

The original contributions presented in this study are included in the article. Further inquiries can be directed to the corresponding author.

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
