# Peer review of "Real-World Data Confirm That the Integration of Deuterium Depletion into Conventional Cancer Therapy Multiplies the Survival Probability of Patients"

_biomedicines, 2025, doi:10.3390/biomedicines13040876_

Round 1
Reviewer 1 Report
Comments and Suggestions for Authors
The authors evaluated the use of deuterium in combination with other conventional cancer therapies. The manuscript is well-written, thought-provoking and quite novel.
The use of deuterium over the decades have shown to have potent anti-cancer therapies and recently shown to be used as an imaging tool in patients with cancer. Deuterium shows promise both for future therapeutic and diagnostic uses.
Minor comment:
Use of deuterium is not without risk. Even though not inherently radioactive it could cause some systemic side effects. This might potentially impact the benefits or advantages it might have over standard anti-cancer therapies. This should be described in the manuscript
Figures and tables are ok. All other sections ok.
Author Response
Reviewer 1.
The authors evaluated the use of deuterium in combination with other conventional cancer therapies. The manuscript is well-written, thought-provoking and quite novel.
The use of deuterium over the decades have shown to have potent anti-cancer therapies and recently shown to be used as an imaging tool in patients with cancer. Deuterium shows promise both for future therapeutic and diagnostic uses.
Minor comment:
Use of deuterium is not without risk. Even though not inherently radioactive it could cause some systemic side effects. This might potentially impact the benefits or advantages it might have over standard anti-cancer therapies. This should be described in the manuscript
Figures and tables are ok. All other sections ok.
Response to Reviewer 1.
Thank you for evaluating the manuscript and for your positive comments.
As deuterium plays an essential role in regulating cell growth and metabolism, it is reasonable to assume that a prolonged deficiency of D may have systemic side effects. This is a reasonable point that we accept, but none of the clinical trials showed any sign of indicating side effects. Likewise, no side effects were experienced by people consuming DDW for years, who were either healthy or cancer patients in remission. In patients with existing tumor, DDW caused side effects, but these were related to tumor necrosis. We described these side effects in the manuscript (line 644-649) as weakness, drowsiness, fever spikes, etc.
On the other hand, regarding the healthy population, it cannot be excluded that a rapid decrease in D concentration may cause side effects. The lowest D concentration we used was 25 ppm, and we did not see side effects. The blood counts of dogs and cats, treated by the veterinary preparation VETERA-DDW-25 (containing 25 ppm D) did not show deterioration either. All the same, we cannot exclude that extreme D depletion (down to 1-10 ppm) may cause adverse effects.

Reviewer 2 Report
Comments and Suggestions for Authors
In this work, the authors investigate the impact of deuterium-depleted water (DDW) on cancer survival, presenting real-world data from 2,649 patients to highlight its potential as an adjunct to conventional therapy. Their findings suggest that integrating DDW could significantly enhance median survival time, warranting further controlled clinical studies. Below are some of my comments:
1) The paper discusses deuterium depletion’s role in cancer treatment but lacks a precise mechanistic explanation at the molecular level. A deeper biochemical analysis would strengthen the argument.
2) The comparison of survival data with historical controls is useful but could be improved by including more global datasets to validate the findings beyond the Hungarian cancer population.
3) The study reports a heterogeneous application of DDW; defining a standardized protocol for optimal dosage and duration would enhance reproducibility.
4) Different cancer types may respond variably to DDW treatment. Further stratification by tumor type, stage, and molecular profile could yield more specific insights.
5) While the study notes that DDW consumption is generally safe, a more detailed longitudinal analysis of potential adverse effects and patient-reported experiences would improve clinical relevance.
6) Since DDW is proposed to affect cellular metabolism, an exploration of metabolic pathways and potential interactions with genetic mutations in different cancers would add depth to the study.
7) More clarity is needed on how DDW interacts with chemotherapy, radiotherapy, and targeted therapies, whether it enhances or inhibits their effectiveness.
8) If possible, add a graphical abstract.
Comments on the Quality of English LanguageMinor improvements required.
Author Response
Reviewer 2.
In this work, the authors investigate the impact of deuterium-depleted water (DDW) on cancer survival, presenting real-world data from 2,649 patients to highlight its potential as an adjunct to conventional therapy. Their findings suggest that integrating DDW could significantly enhance median survival time, warranting further controlled clinical studies. Below are some of my comments:
- The paper discusses deuterium depletion’s role in cancer treatment but lacks a precise mechanistic explanation at the molecular level. A deeper biochemical analysis would strengthen the argument.
Thank you for evaluating the manuscript, for your comments, and for your suggestions.
Based on the available data, our conclusion is that the D/H ratio and its changes exert a moderate effect on all biochemical processes simultaneously and not a significant effect on one or two specific biochemical processes. This is supported by the fact that, due to the isotopic effect, it can be assumed that all chemical reactions are slower with D. In this MS, we wanted to focus on describing and explaining the effect of the D/H ratio on cancer-related genes and tyrosine kinase genes. As these genes play a key role in regulating the cell cycle, and the gene expression study showed that a higher D/H ratio stimulates the expression of these genes, we apparently could demonstrate an element of the mechanism of action of D depletion. (line 601-622)
The “deeper biochemical analysis” mentioned by the reviewer has indeed to be done in the future. We cannot exclude that, by continuing the research, a single protein with high efficacy in collecting deuterium and playing a very significant role in this sub-molecular regulatory system will be identified.
- The comparison of survival data with historical controls is useful but could be improved by including more global datasets to validate the findings beyond the Hungarian cancer population.
In the manuscript, we cited the two most reliable datasets from the US and the EU. Even these two datasets indicate differences in the efficacy of cancer treatment, as in the EU, 40% is the death rate compared to the incidence rate, while in the US it is only 31.5%. This ratio is 47.1%, in Hungary, indicating the leading position of the country in cancer mortality. The real-world data indicate that integrating deuterium depletion into existing cancer therapy can with an MST of 11.6 years, Hungary's annual cancer death toll could drop from 33,000 to approximately 5,000–7,000, potentially saving 26,000–28,000 lives each year. which is a significant improvement and is obvious without the inclusion of more comprehensive global datasets. Also, when calculating with MSTs, it is not much longer in the European and American cancer population than among Hungarians (line 237). So, the estimated MST benefit would be nearly the same when using non-Hungarian historical control data.
- The study reports a heterogeneous application of DDW; defining a standardized protocol for optimal dosage and duration would enhance reproducibility.
The heterogeneity of the DDW-consuming cancer population requires various protocols to handle the diversity. In a book by G. Somlyai referred to in the MS (ref. 24) detailed protocols for patients suffering from various malignancies in different stages are described, together with optimal ways to integrate DDW to surgery, radiotherapy, hormone therapy, and chemotherapy. However, it seemed inadequate to include such protocols in the text of this MS.
- Different cancer types may respond variably to DDW treatment. Further stratification by tumor type, stage, and molecular profile could yield more specific insights.
To show the variability of the effect of DDW in different cancer types, we added the calculated and published MST in the case of lung, breast, and glioblastoma cancer (line 99-100). Detailed stratification, that the data set allows, was not within the scope of the present work but shall be done in the future.
- While the study notes that DDW consumption is generally safe, a more detailed longitudinal analysis of potential adverse effects and patient-reported experiences would improve clinical relevance.
As deuterium plays an essential role in regulating cell growth and metabolism, it is reasonable to assume that a prolonged deficiency of D may have systemic side effects. This is a reasonable point that we accept, but none of the clinical trials showed any sign of indicating side effects. Likewise, no side effects were experienced by people consuming DDW for years, who were either healthy or cancer patients in remission. In patients with existing tumor, DDW caused side effects, but these were related to tumor necrosis. We described these side effects in the manuscript (line 644-649) as weakness, drowsiness, fever spikes, etc.
On the other hand, regarding the healthy population, it cannot be excluded that a rapid decrease in D concentration may cause side effects. The lowest D concentration we used was 25 ppm, and we did not see side effects. The blood counts of dogs and cats, treated by the veterinary preparation VETERA-DDW-25 (containing 25 ppm D) did not show deterioration either. All the same, we cannot exclude that extreme D depletion (down to 1-10 ppm) may cause adverse effects.
- Since DDW is proposed to affect cellular metabolism, an exploration of metabolic pathways and potential interactions with genetic mutations in different cancers would add depth to the study.
We agree that it is necessary to explore the connection between genetic mutations and the efficacy of deuterium depletion. In the manuscript, two formerly published papers are referred to (ref. 25 and 27) where we prove that decreased D concentration resulted in lower gene expression of c-Myc, K-Ras, Bcl2, oncogenes involved in carcinogenesis. However, much further research will be necessary to identify gene by gene and tumor by tumor the possible role of a given gene in the development of certain tumors (and the particular action of D-depletion on that).
- More clarity is needed on how DDW interacts with chemotherapy, radiotherapy, and targeted therapies, whether it enhances or inhibits their effectiveness.
Although our study evaluated the data retrospectively, it was evident that combining DDW consumption with radiotherapy yielded better results than that expected from radiotherapy alone. We added two new sentences to the manuscript to emphasize this positive interaction. (Line 627-634). As to other conventional therapies, a positive interaction between any two treatments that impede the survival of cancer cells is obvious or at least expectable.
- If possible, add a graphical abstract.
We suggest postponing the graphical abstract for now.

Reviewer 3 Report
Comments and Suggestions for Authors
The manuscript addresses a important topic that conversion of deuterium to hydrogen increases the survival of the cancer patients. The authors have taken the data from Hungarian population and shown that lower deuterium has better survival chances. The analysis of the data is nicely and robustly done. However there are few important points which need to be discussed.
- Is it even possible to control the levels of Deuterium in water sources? Measuring deuterium activity in all water samples will be highly expensive.
- Can the authors also take some data from non Hungarian population say where it is known that deuterium levels are low or high and compare these results with that. They might get some better insights.
Author Response
Reviewer 3.
The manuscript addresses a important topic that conversion of deuterium to hydrogen increases the survival of the cancer patients. The authors have taken the data from Hungarian population and shown that lower deuterium has better survival chances. The analysis of the data is nicely and robustly done. However there are few important points which need to be discussed.
- Is it even possible to control the levels of Deuterium in water sources? Measuring deuterium activity in all water samples will be highly expensive.
Thank you for evaluating the manuscript and for your questions.
There are data published on D concentration of the water at various places of the planet. Near the equator, D concentration of the water is approximately 150-155 ppm. As we move toward to the poles, D concentration decreases; in Norway, it could be as low as 135 ppm (but even this is higher than the D concentration of the least depleted drinking water product, Preventa 125, that is by the way recommended for healthy people for health preservation, not for tumor patients as a cure). We did not take any such measurement on our own, it is not within the scope of our research. As to any relationship with cancer prevalence, see our response to the next question.
- Can the authors also take some data from non Hungarian population say where it is known that deuterium levels are low or high and compare these results with that. They might get some better insights.
As numerous factors influence the health of a given population, including the occurrence of cancer, it seems complicated to establish any clear correlation between D concentration of the local water sources and incidence of cancer. All the same, an example where correlation between the D concentration and cancer incidence may exist could be that of the people living in the Caucasus, where high altitude and distance from the ocean result in a low D concentration of surface waters, and cancer incidence is low, so the population has long healthy lifetime.

Round 2
Reviewer 2 Report
Comments and Suggestions for Authors
Authors thoroughly addressed my concerns.